# REDfly: An Integrated Knowledgebase for Insect Regulatory Genomics

**DOI:** 10.3390/insects13070618

**Published:** 2022-07-11

**Authors:** Soile V. E. Keränen, Angel Villahoz-Baleta, Andrew E. Bruno, Marc S. Halfon

**Affiliations:** 1Independent Research, Berkeley, CA 94705, USA; soileredfly@gmail.com; 2Center for Computational Research, State University of New York at Buffalo, Buffalo, NY 14203, USA; angel.villahoz-baleta@usda.gov (A.V.-B.); aebruno2@buffalo.edu (A.E.B.); 3New York State Center of Excellence in Bioinformatics and Life Sciences, State University of New York at Buffalo, Buffalo, NY 14203, USA; 4Department of Biochemistry, State University of New York at Buffalo, Buffalo, NY 14203, USA; 5Department of Biomedical Informatics, State University of New York at Buffalo, Buffalo, NY 14203, USA; 6Department of Biological Sciences, State University of New York at Buffalo, Buffalo, NY 14203, USA; 7Department of Molecular and Cellular Biology and Program in Cancer Genetics, Roswell Park Cancer Institute, Buffalo, NY 14263, USA

**Keywords:** insects, *Drosophila*, regulatory genomics, gene regulation, *cis*-regulatory module, enhancer, genome annotation

## Abstract

**Simple Summary:**

Understanding how genes are regulated is a vital area of current biological research and a crucial adjunct to ongoing efforts to sequence entire genomes. Knowing the DNA sequences responsible for gene regulation—transcriptional *cis*-regulatory modules (CRMs, e.g., “enhancers”) and transcription factor binding sites (TFBSs)—is important for many areas of research including interpretation and validation of data developed by large-scale genomics projects, providing training data for machine-learning CRM-discovery methods, genome annotation, modeling gene-regulatory networks, studying the evolution of gene regulation, and numerous aspects of the basic biology of transcriptional regulation. Knowledge of insect CRMs is also an important step in developing biotechnology methods for control of insect disease vectors and for eliminating pathogen transmission. The REDfly (Regulatory Element Database for Fly) database integrates all of the available insect *cis*-regulatory information from multiple sources to provide a comprehensive collection of known regulatory elements. In this paper, we describe REDfly’s basic contents and data model, emphasizing recently added features, and provide illustrated walk-throughs of some common search scenarios.

**Abstract:**

We provide here an updated description of the REDfly (Regulatory Element Database for Fly) database of transcriptional regulatory elements, a unique resource that provides regulatory annotation for the genome of *Drosophila* and other insects. The genomic sequences regulating insect gene expression—transcriptional *cis*-regulatory modules (CRMs, e.g., “enhancers”) and transcription factor binding sites (TFBSs)—are not currently curated by any other major database resources. However, knowledge of such sequences is important, as CRMs play critical roles with respect to disease as well as normal development, phenotypic variation, and evolution. Characterized CRMs also provide useful tools for both basic and applied research, including developing methods for insect control. REDfly, which is the most detailed existing platform for metazoan regulatory-element annotation, includes over 40,000 experimentally verified CRMs and TFBSs along with their DNA sequences, their associated genes, and the expression patterns they direct. Here, we briefly describe REDfly’s contents and data model, with an emphasis on the new features implemented since 2020. We then provide an illustrated walk-through of several common REDfly search use cases.

## 1. Introduction

The turn-of-the-century advent of fully sequenced metazoan genomes brought with it the first genome annotations, which were largely confined to positions of confirmed and predicted genes, and typically housed in community-specific model-organism databases, e.g., [1,2,3,4]. Remarkably, over two decades later, the major databases (see [5]) are still mostly lacking annotation of non-coding regulatory sequences. These sequences include distal “*cis*-regulatory modules” (CRMs), a generic term encompassing such regulatory elements as enhancers, which mediate positive gene regulation; silencers, involved in negative regulation; and a growing number of additional elements that are not easily classified including PREs, super-enhancers, insulators, tethering elements, and others [6,7,8,9,10,11].

Obtaining a comprehensive annotation of regulatory sequences is important not only for its intrinsic value in illuminating the structure and function of the genome, but also for its practical value in facilitating bioinformatics analyses of CRMs and their interactions with other genomic features. To this end, the REDfly database (Regulatory Element Database for fly) [12,13,14,15] plays a critical role for regulatory bioinformatics and genomics, particularly with respect to insects. REDfly is a highly curated knowledgebase dedicated to annotating and integrating the growing body of information on insect transcriptional regulatory sequences curated from the published literature, with an emphasis on empirically validated CRMs. Although originally focused solely on *Drosophila melanogaster*, REDfly now includes data from a growing number of additional insect species.

## 2. Utility of REDfly

Prior to the development of REDfly, large-scale analyses of regulatory sequences were challenging to conduct, as the bulk of the existing regulatory data was distributed among hundreds of individual publications. Consequently, what few analyses were completed were performed on small and frequently biased sets of CRMs, such as a limited subset of early developmental pair-rule stripe enhancers in *Drosophila*, e.g., [16,17,18]. By curating these data and making them findable, accessible, interoperable, and usable (FAIR) [19], REDfly made it possible to bring statistical, computational, and comparative genomics methods to bear on their study. REDfly enabled the first-ever large-scale, relatively unbiased analysis of CRMs, which immediately revealed novel insights into CRM-sequence composition, differences among tissue-specific groups of CRMs, and an early indication of the presence of enhancer RNAs (eRNAs) as a prevalent CRM characteristic [20]. REDfly, by continuing to compile the data from hundreds and eventually thousands of individual experiments scattered throughout four decades of literature, subsequently proved instrumental in facilitating studies in a wide variety of research areas, including:

*Biology of CRMs*. REDfly has been used to investigate the organization of TFBSs within CRMs [21] and how combinatorial binding influences CRM activity [22]. Soluri et al. [23] investigated how pioneer TFs control chromatin accessibility, and Blick et al. [24] examined the ability of CRMs to act in *trans*. REDfly data helped to illustrate how CRMs can have multiple functions [25], such as dual use as both enhancers and Polycomb response elements [26], or as both enhancers and silencers [27].

*Interpretation of genomic data*. REDfly has been critical for interpreting data from large-scale genomics projects including TF binding studies, e.g., [28,29] and studies of insulators [30,31]. A study challenging our understanding of which epigenetic marks characterize regulatory sequences depended on REDfly data [32]. REDfly has been used to study chromosome domains and chromatin “states”, e.g., [33,34,35,36], to explore 3D-chromatin conformation [37,38], to study ncRNA and eRNA expression [39,40], and to validate scATAC-seq approaches, e.g., [41,42].

*Computational CRM discovery*. REDfly has played a dramatic role in methods for computational CRM discovery, both as a source of training data and as a method for validating predictions, e.g., [43,44,45,46,47,48,49,50,51]. Su et al. [52] used REDfly data to assess CRM-discovery approaches, which would have been impossible without REDfly. Computational CRM-discovery methods using REDfly for training data also can identify CRMs in diverse insect species [53] and, as such, provide a powerful tool for annotating insect regulatory genomes [54].

*CRM evolution*. REDfly has enabled studies of CRM evolution and TFBS turnover, e.g., [55,56,57,58,59,60,61]. Wang et al. [62] used REDfly data to investigate the selective pressure on DNA shape at TF binding sites, and Peng et al. [63] explored the relationship between chromatin accessibility and TF binding to predict evolutionary changes in enhancer activity.

As can be seen from these examples, REDfly is an important source of raw data for analysis, hypothesis generation, assessment, validation, and empirical research. In the remainder of the paper, we describe the REDfly data model and provide a guide to some common REDfly uses.

## 3. REDfly Data Model

REDfly curates two types of data: CRMs and transcription factor binding sites (TFBSs), with CRMs being the main focus. Historically, CRM annotations have been drawn from reporter gene assays in transgenic animals or cultured cells, but an increasingly diverse set of assays are starting to be included. In particular, several years ago REDfly started to capture CRMs identified through various “X-seq” assays, such as ATAC-seq, FAIRE-seq, DNase-seq, ChIP-seq, etc., as well as from purely computational predictions, in recognition of the fact that many regulatory sequences are presently being defined by these methods. There is considerable debate in the regulatory genomics field as to just how CRMs should be defined, with several studies indicating that the different methods of CRM identification have led to widely non-overlapping results and raising questions as to which, if any, of these methods most accurately identify CRMs [64,65,66]. As a result, REDfly separates its CRM data into four distinct subclasses: *reporter constructs (RC)*, *CRM_segments*, *predicted CRMs (pCRM)*, and *inferred CRMs (iCRM)*. *RCs* and *CRM_segments* are drawn from activity-based assays (Figure 1, left), primarily gene-expression data from either reporter genes or from native genes following mutation or deletion of regulatory sequences. *RCs* mainly represent reporter-gene results, assayed either in transgenic animals or in cell-culture assays; the two types of assays can be independently searched. The *CRM_segment* class contains sequences that are demonstrated to be necessary for gene regulation but, unlike in a reporter gene assay, are not necessarily sufficient. Such sequences can be obtained from the analysis of small chromosomal deletions or site-directed mutagenesis but are increasingly being found in the literature as a result of CRISPR/Cas9-mediated targeted sequence deletions. *pCRMs*, on the other hand, reflect CRMs identified by assays that do not require demonstration of activity, for example, the presence of histone modifications, or computational predictions (Figure 1, right). *iCRMs* are not curated from the literature, but represent putative regulatory elements based on the analysis of other REDfly data (see below).

### 3.1. “Reporter Constructs” and Their Attributes

*RCs* in REDfly have three primary associated attributes: *expression*, *CRM*, and *minimization* (Figure 2). *Expression* can be either “positive” or “negative” and denotes whether or not the sequence drives gene expression in the reporter-gene assay. Positive expression is described using the *Drosophila* Anatomy Ontology [67] (or, for non-*Drosophila* species, an appropriate species-specific anatomy ontology). Every expression pattern is linked with the stage(s) of development at which this expression is observed, using community-standard developmental staging ontologies, and with an indicator for whether that expression is consistent with or ectopic to that of the assigned target gene. Sexually dimorphic expression is also captured. Terms from the Gene Ontology [68] allow for annotation of regulatory elements that respond to specific signals or environmental cues (e.g., wound healing, hypoxia, circadian cycling).

Recent data suggest that many regulatory sequences can act as enhancers in one cell type, while simultaneously acting as transcriptional silencers in another [27]. An “enhancer/silencer” tag associated with each described expression pattern allows for this dual functionality to be accurately represented (Figure 2). For example, [69] describe a sequence, *Tm2_intronI1B(b)*, that acts as a silencer by suppressing activity of the *Tm2_intronI1b(a)* enhancer in embryonic and larval muscle, but which also acts as a weak enhancer to promote expression in adult leg muscle. In the REDfly “anatomical expression” tab, this *RC* would, therefore, have the annotation “embryonic/larval somatic muscle” and “silencer” in one row, and “skeletal muscle of leg” and “enhancer” in another.

The *CRM* attribute indicates whether an *RC* is the shortest of a set of nested sequences with identical activity, i.e., what is commonly referred to as a “minimal enhancer” (Figure 3B,D,F). When an *RC* is part of a set of nested sequences, we say the set of *RCs* has undergone *minimization* (Figure 3C–I). This attribute is included as an aid to researchers to help decide whether to undertake experimental analysis of a region, as a minimized region might provide less new information than one for which only a single construct has been tested. Note that none of these attributes are fixed for a given RC record, as the attribute values might change over time as new information becomes available through follow-up studies and new literature.

### 3.2. “Inferred CRMs”

*iCRMs* represent sequences that have not been explicitly identified as CRMs from any assay, but that are inferred to have regulatory activity based on analysis of other sequences in REDfly. For instance, overlapping sequences may have the same regulatory activity when assayed in vivo, and a logical—although unproven—supposition in such a case is that the overlapping region contains the “true” minimal CRM (Figure 3I). These overlaps can arise from *RCs* that were assayed in different publications and, therefore, are only discovered through integrated curation in REDfly.

## 4. Species in REDfly

Although REDfly has historically been focused on *Drosophila melanogaster*, the clear value of comparative genomics and of working with non-traditional model organisms, a vast increase in the number of sequenced insect genomes, and the small but growing availability of both predicted and validated insect regulatory sequences has led us to expand REDfly by implementing the ability to curate regulatory sequences for additional insect species. Information on which species are represented can be found on the “Species” page; current species include the mosquitoes *Anopheles gambiae* and *Aedes aegypti* and the beetle *Tribolium castaneum*. Additional species will be added as data accumulate. Since insect CRMs are often tested using transgenic *Drosophila*, REDfly divides the sequence and gene-feature data and the expression pattern and cell-line data into separate components. Each REDfly record has both a “sequence from” and an “assayed in” component. Sequence and gene-feature data are linked to the former, and anatomy and staging data to the latter. While it is preferable to describe species-specific reporter-gene-expression patterns using the proper species-specific anatomy ontology, many species lack an ontology as rich in terms as that for *Drosophila*. Therefore, terms from the *Drosophila* ontology can also be used to annotate expression in other species.

In order to facilitate research using these newly added genomes, we have implemented interfaces for *BLAT* [70] and in silico *PCR* [71] for each species included in REDfly. These can be accessed through the “Species” page.

## 5. Contents of REDfly

REDfly has continued to expand its contents at a rapid rate (Table 1). Since the end of 2019, the number of curated publications has increased by 30%, leading to an increase in the total number of Reporter Construct records by over 25% and in the number of pCRMs by almost 60%. Not reflected in these numbers, however, is an ambitious endeavor to update all RC records with the full set of RC expression attributes (developmental staging, sexually dimorphic activity, ectopic activity, and enhancer/silencer activity), which did not become a full part of the REDfly data model until the release of REDfly v6 in 2020. Since that time, over one-third of the RC records have been updated to contain this information.

## 6. Using REDfly

The extensive data and metadata REDfly provides for each of its records allows for detailed customized searching of the database contents. Typical entry points for a REDfly search are via a gene name (Figure 4A(a)) or a literature reference (via PubMed ID, Figure 4A(b)). By default, searching for a gene name will execute a “by locus” search in which any elements annotated as being associated with that gene, as well as any elements found within a user-customizable range of 10 kb upstream or downstream of the gene (regardless of assigned target gene), will be returned. Moreover, by default, elements identified solely by assays performed in cultured cells are omitted from the results (Figure 4A(d)); unchecking the check-box causes these results to be included.

Clicking on the “Advanced Search” arrow (Figure 4A(e)) allows access to a large variety of additional options (Figure 4B), including the ability to restrict searches to specific *RC* attributes, genomic locations, anatomical regions, developmental stages, or biological processes. More detailed and complex search capabilities are under development. 

Regardless of whether “basic” or “advanced” search is used, a summary of the results will appear in the “Search Results” pane directly below the main search window (Figure 4A(g)). Results for each REDfly data class—*RCs/CRMs*, *CRM_segments*, *pCRMs*, *TFBSs*, and *iCRMs*—are displayed in individual tabs to make it easier to view results by type. Checkboxes allow selection of records for download (Figure 4A(h)) in any of a number of convenient formats. Alternatively, clicking on an individual result will open a multipaned “Detailed View” window containing full information for the selected record (Figure 4C). Basic location and attribute data are displayed in the “Information” tab, along with links to relevant model-organism databases and genome browsers (Figure 4C). The “Location” tab provides a snapshot of the element in its genomic milieu (Figure 4D). The “Sequence” tab displays the genomic sequence and its size (Figure 4E), while the “Citation” tab (Figure 4F) provides a citation and link to the publication describing the current element, plus a description of the evidence used by REDfly curators to annotate sequence and expression information. The “Anatomical Expression” tab (Figure 4G) lists each cell type or tissue where the regulatory element is active, along with a specific citation for that activity data (since activity data may be drawn from multiple references), developmental staging for the observed activity, and the other attributes discussed above, e.g., sexually dimorphic activity, ectopic expression, and enhancer or silencer activity.

Since CRM activity can be complex and not easily summarized using the anatomical and staging terms available in the relevant ontologies, we also supply a “Notes” tab containing details and clarifications.

## 7. Use Cases

Below, we illustrate several common scenarios for using REDfly, with step-by-step instructions.

*Note*: The “Clear Search Fields” button (Figure 4A(f)) can be used to reset the search interface to the default settings and empty all search fields.

Case 1: I Want to Find All *D. melanogaster* Regulatory Features within a Specified Locus.

One common use of REDfly is to explore what is known about the regulation of a particular gene. This is easily done:(a)Make sure that both the “sequence from” and “assayed in” fields (Figure 4A(c)) are set to *Drosophila melanogaster*;(b)Select the locus of interest using the “Gene Name” field (Figure 4A(a));(c)Make sure the radio button is set to “by locus”;(d)Click on “search” (Figure 4A(d)).

This will retrieve all features directly associated with the specified gene as well as any other features within 10 kb upstream or downstream of that gene. To alter the size of the region to be searched do the following:(e)Open the Advanced Search box (Figure 4A(e));(f)Change the value in the “Search Range Interval (−/+)” field from 10,000 to the desired number of basepairs;(g)Click on “search” (Figure 4B(o)).

Case 2: I Want to Find All *D. melanogaster* Regulatory Features within a Genomic Region.

It is also simple to determine what regulatory features are present in a given region of the genome, of arbitrary size, without needing to specify a particular gene or genes within the region:(a)Make sure that both the “sequence from” and “assayed in” fields (Figure 4A(c)) are set to *Drosophila melanogaster*;(b)Open the Advanced Search box (Figure 4A(e));(c)Set the “Chromosome”, “Start Coord.”, and “End Coord.” fields (Figure 4B(j)) to reflect your region of interest (make sure that the chromosomes are from the correct species, e.g., “3R (dmel)”;(d)Click on “search” (Figure 4B(o)).

Case 3: I Want to Find All *Anopheles gambiae* Sequences Tested for Regulatory Activity Using a Transgenic Anopheles Gambiae Assay.

As REDfly begins to curate regulatory data from a wider variety of species, it become important to be able to isolate data for a species of interest, as well as to be able to separate out data obtained by direct observation in the desired species, versus heterologous testing in a host organism or cells of a different species. The two “species” search fields allow for these distinctions.(a)Set the “sequence from” and “assayed in” fields (Figure 4A(c)) to “*Anopheles gambiae*”;(b)Click on “search” (Figure 4A(d)).

Case 4: I Want to Find All *Aedes aegypti* Sequences Tested for Regulatory Activity Using a Transgenic Drosophila Assay.

(a)Set the “sequence from” field (Figure 4A(c)) to “*Aedes aegypti*”;(b)Set the “assayed in” field “*Drosophila melanogaster*”;(c)Click on “search” (Figure 4A(d)).

Case 5: I Want to Find All *D. melanogaster* Sequences That Were Tested In Vivo and Found to Be Negative for Regulatory Activity.

It is often desirable to know that a sequence has been tested for regulatory activity and demonstrated not to be a CRM, at least in a particular context. Since REDfly curates all experimental data, regardless of outcome, such a search is straightforward.(a)Make sure that both the “sequence from” and “assayed in” fields (Figure 4A(c)) are set to *Drosophila melanogaster*;(b)Make sure the “Exclude Cell Culture Only” box is checked (Figure 4A(d));(c)Open the Advanced Search box (Figure 4A(e));(d)In the “Restrictions” section (Figure 4B(i)), check the “Negative Expression Only” box;(e)Click on “search” (Figure 4B(o)).

Case 6: I Want to Find All *D. melanogaster* Regulatory Sequences Shorter Than 1 kb in Length Discovered Using Reporter Gene Assays in Cell Culture (Excluding Results from STARR-Seq Assays).

Investigators test sequences of greatly varying size for regulatory activity, but, for many uses, one might want to focus on sequences of only a certain length. We show how to do that in this use case, along with an illustration of how to confine results to those obtained in cell culture reporter gene assays other than STARR-seq [72].(a)Make sure that both the “sequence from” and “assayed in” fields (Figure 4A(c)) are set to *Drosophila melanogaster*;(b)Uncheck the “Exclude Cell Culture” box (Figure 4A(d));(c)Open the Advanced Search box (Figure 4A(e));(d)Type “1000” in the “Maximum Size” box (Figure 4B(k));(e)In the “Restrict evidence to” dropdown select “reporter construct (cell culture)” (Figure 4B(l));(f)Click on “search” (Figure 4B(o)).

Case 7: I Want to Find All *D. melanogaster* Regulatory Sequences That Drive Reporter Gene Expression in the Larval Wing Disc as a Response to Injury.

Some of the most powerful uses for REDfly data are to assemble tissue-specific or function-specific sets of CRMs for experimental use or bioinformatic analysis. The use of anatomy, staging, and biological process ontologies provides for great flexibility and allows the user to determine how granular of a search to conduct.(a)Make sure that both the “sequence from” and “assayed in” fields (Figure 4A(c)) are set to *Drosophila melanogaster*;(b)Open the Advanced Search box (Figure 4A(e));(c)In the “Anatomical Expression Term” box (Figure 4B(m)), begin typing “wing disc” and select this term when it appears in the completion list;(d)In the “Biological Process Term” box (Figure 4B(n)), begin typing “wound healing” and select “Response to wounding” from the completion list;(e)Click on “search” (Figure 4B(o)).

The above use cases illustrate only some of the many types of ways to access the REDfly regulatory data. Moreover, as powerful as these search capabilities are, they have not fully kept up with the growth of REDfly’s data model over the years, such that the data contained in REDfly are in fact richer than what can be easily searched and downloaded. A major development priority over the next year will be to introduce new and more flexible search and download capabilities along with improved integration of the different data types and species curated by REDfly. We also expect there to be considerable growth in several data categories. These include *CRM_segments*, as researchers increasingly turn to CRISPR/Cas9-mediated deletion of regulatory sequences, and *pCRMs* from multiple new species as experimental methods, such as single-cell ATAC-seq [73] and computational CRM discovery methods, such as SCRMshaw (reviewed by [54]) continue to be applied to sequenced insect species at a rapid rate.

## 8. Access to REDfly

REDfly is freely available to the public at http://redfly.ccr.buffalo.edu. News about database updates and new features can be obtained from our Twitter feed @REDfly_database. Subsets of REDfly data can also be obtained from FlyBase [74] and FlyMine [75].

## 9. Data Submission

REDfly is a curated resource, with all data entry handled by our team of biocurators. The biocuration team engages in frequent back-and-forth communication with authors to ensure that published regulatory data are accurately reflected in REDfly’s records. This includes a post-curation author review step, through which an e-mail is automatically sent inviting the corresponding author of each newly curated paper to review and, if necessary, correct the REDfly annotation. Although there is currently no mechanism for direct outside data submission, we encourage members of the community to use the “contact us” function on the website to alert us to missing or incorrect annotations and to work with us on making sure their data are included in REDfly in a timely manner.

## Figures and Tables

**Figure 1 insects-13-00618-f001:**
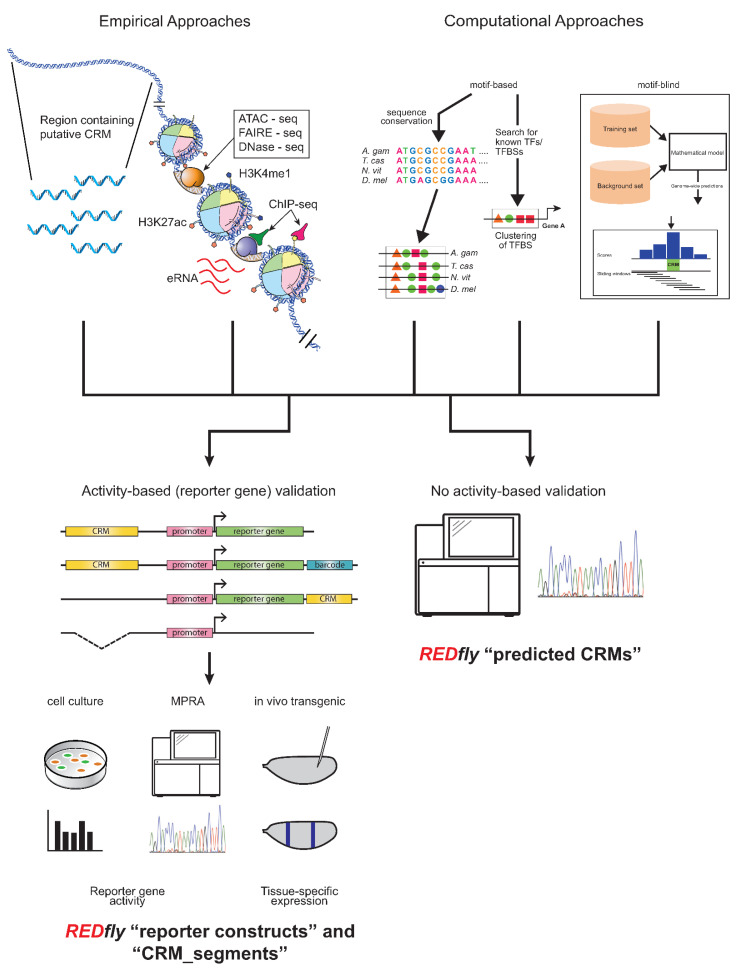
Activity-based and non-activity-based methods for defining regulatory sequences. Top: A wide variety of methods exist for identifying regulatory sequences based on both empirical (**left**) and computational (**right**) approaches. Empirical approaches include unbiased testing of non-coding DNA regions as well as selection of sequences based on chromatin accessibility, histone post-translational modification, transcription factor binding, production of enhancer RNAs, and others. Computational approaches may include assessment of sequence conservation, presence of transcription factor binding motifs, or various machine-learning methods. Bottom: Results from these regulatory element discovery methods can be obtained with (**left**) or without (**right**) the use of activity-based criteria. Activity-based criteria typically involve some sort of reporter-gene assay, which might be performed in cultured cells, using next-generation sequencing in a “massively parallel reporter assay” (MPRA), or in transgenic animals; recently, testing via genomic deletion via CRISPR/Cas9 has also been gaining popularity. REDfly classifies regulatory sequences derived from these methods as *reporter constructs (RCs)* and *CRM_segments*, while any methods that identify regulatory sequences without recourse to activity-based criteria are referred to as *predicted CRMs (pCRMs)*. These somewhat historical definitions should not be construed to imply that one or the other type of data is more accurate or “correct”.

**Figure 2 insects-13-00618-f002:**
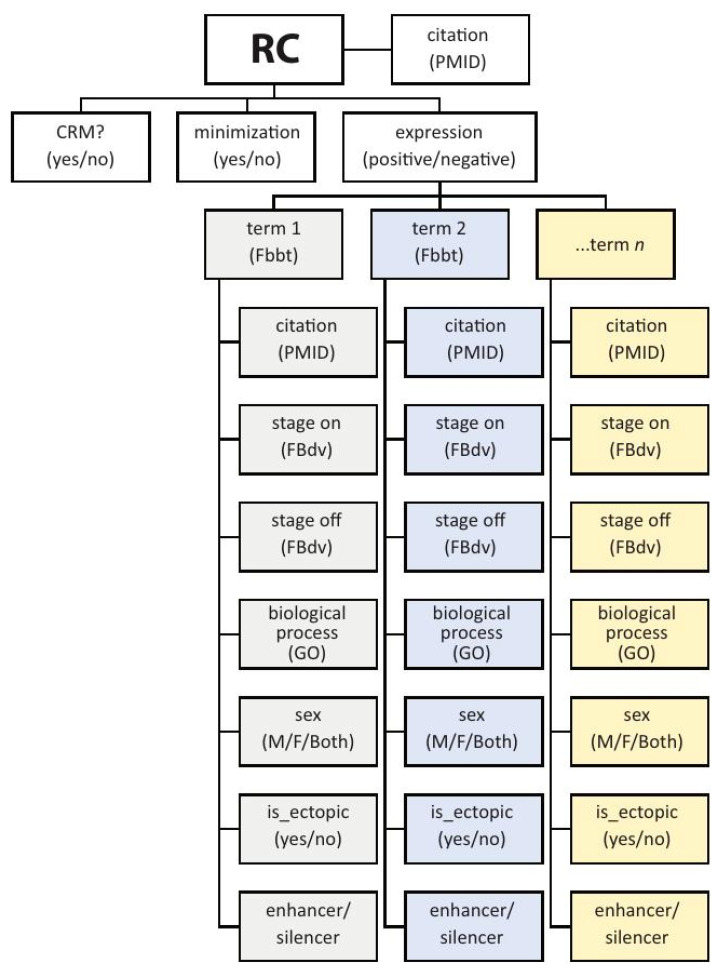
The basic REDfly *Reporter Construct* data model. Depicted is a partial illustration of the data model for REDfly *Reporter Construct (RC)* records. Each RC has the three basic attributes of *CRM*, *minimization*, and *expression*. If *expression* is positive, the RC is annotated with each anatomical location where expression is observed, to the most granular degree available, based on a species-relevant anatomy ontology. Each annotated expression pattern is then associated with additional data including citation; the stages at which expression is observed; a biological process (where relevant) drawn from the Gene Ontology; the sex in which the expression is observed; whether or not the expression is ectopic with respect to the known pattern of the associated gene; and whether the RC is acting as an enhancer or a silencer in the current tissue. Not included in the schematic are additional RC-associated data such as species, gene names, relevant figure panels, evidence terms, and others.

**Figure 3 insects-13-00618-f003:**
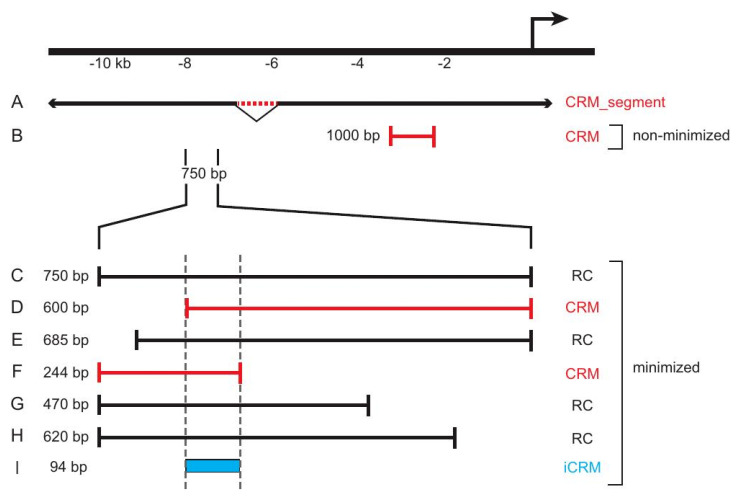
REDfly data subclasses and *Reporter Construct* attributes. The figure illustrates a hypothetical locus for which different sequences have been tested in vivo. Sequence (**A**) represents an approximately 1 kb genomic deletion (red dotted line) that reduces expression from the nearby promoter 6 kb downstream (bent arrow). As such, it is considered a *CRM_segment*. Sequence (**B**) is a 1 kb sequence fragment located roughly 2 kb upstream of the transcription start. Since it is an isolated sequence, it is considered to be a *CRM* that has not been subject to minimization. If this construct showed reporter gene activity, it would be designated as “expression positive”; otherwise, it would be labeled “expression negative.” Constructs (**C**–**H**) are part of an overlapping and partially nested series of sequences spanning 750 bp of DNA 7.25 kb upstream of the transcription start. In this example, each drives the identical pattern of reporter gene expression. Since each of these sequences overlaps at least one other, we consider this region and the six sequences to have undergone minimization. Sequences (**D**,**F**) are each the shortest of a respective set of fully nested sequences and are, therefore, considered to be *CRMs* (marked in red). The remaining sequences are designated as *RCs* (black). A 94 bp sequence, (**I**), marks the minimal region of overlap among all of the sequences and is, thus, registered in REDfly as an *inferred CRM* (*iCRM*, blue).

**Figure 4 insects-13-00618-f004:**
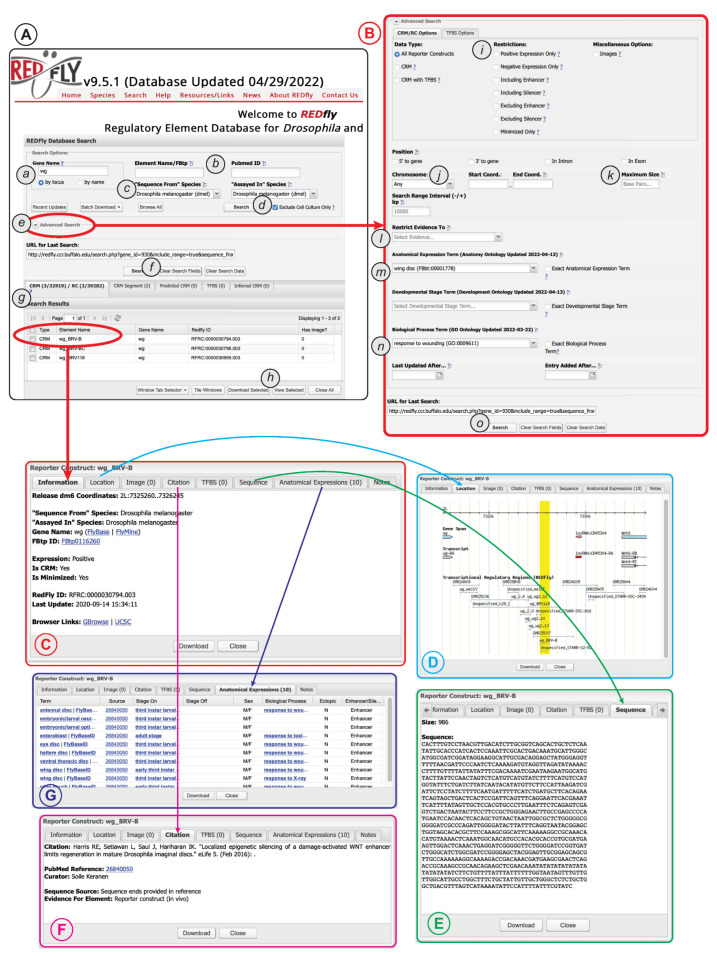
The REDfly search interface. See text for details. (**A**) The basic search panel. (**B**) The Advanced Search panel. (**C**) The Detailed Results “Information” pane. (**D**) The Detailed Results “Location” pane. (**E**) The Detailed Results “Sequence” pane. (**F**) The Detailed Results “Citation” pane. (**G**) The Detailed Results “Anatomical Expression” pane.

**Table 1 insects-13-00618-t001:** REDfly contents as of 1 July 2022.

Reporter Constructs (RCs)	43,819
*From* in vivo *reporter genes*	*21*,*690*
*Associated with staging and other attributes*	*17*,*961*
CRM_segments	16
Predicted CRMs (pCRMs)	14,318
Inferred CRMs (iCRMs)	7760
Transcription Factor Binding Sites (TFBS)	2717
Publications curated	1366

## Data Availability

The data presented in this study are publicly available at http://redfly.ccr.buffalo.edu and as a permanent archive in the University at Buffalo Institutional Repository at http://hdl.handle.net/10477/82107.

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
