# Peer review of "REDfly: An Integrated Knowledgebase for Insect Regulatory Genomics"

_insects, 2022, doi:10.3390/insects13070618_

Round 1
Reviewer 1 Report
This review presents the current state of the REDfly database, an endeavor from the Halfon lab at the University of Buffalo that curates regulatory information primarily for Drosophila melanogaster. The authors begin with introduction to the database and an excellent history of the ways REDfly has been used in the publication record. The authors then describe in detail the different classes and data models present in REDfly, including information on reporter constructs and CRMs, TFBSs, and predicted or inferred CRMs. They end with a walkthrough of example data as viewed through the database's website, and guide the reader through several case studies to search the database for specific questions. I have used the database several times, and the walkthrough does an admirable job of presenting the information for new users. REDfly is an important tool for many Drosophila researchers, and as new species are added, it will grow in value to a broader community. As such, I find this review to be suitable for publication, particularly since there is currently a seemingly exponential growth in the data in this field as of late. I have only one comment, regarding Table 1, I found the formatting confusing - "RC" looks as one would expect for a title or header, but I do not think it is meant to be so. Then, I believe that the two italicized row below "RC" are meant to be subsets within the set of RCs, and the remaining rows are independent of RCs? Again, I am confused, but this should be an easy thing to fix or explain.
Author Response
We thank the reviewer for their kind comments about our work and our manuscript. When we looked at Table 1, we were confused also! It appears that an editorial formatting error before the paper was sent for review led to the confusing formatting. We have restored the Table to its correct appearance, which will hopefully now be readily interpretable.
Reviewer 2 Report
This is a well-written and clear review that outlines important updates to RedFLY, a well-established interactive database of cis regulatory elements (CRMs), focused on Drosophila. The manuscript outlines the significance of the database, and both how it has been updated with newly published information and how it has been extended to include data from novel methods of CRM detection. The manuscript gives a good overview of the interactive pages that are present and gives several useful case examples of how to use these tools to interrogate the data available on the site.
I particularly liked the description of how data is assigned to categories of CRMs outlined in figure 1 - this neatly organizes the source of the data that a user explores using this tool. I was confused as to how silencer elements are categorized - beginning with how they are detected and continuing with how they are assigned a subclass and fit into the ontology matrix shown in Fig. 2. In particular, lines 169-172 should benefit from a specific example of a silencer that was detected withe experimentally or computationally and how that information was fit into the data matrix.
Author Response
We thank reviewer 2 for their kind comments. Like many regulatory elements, there are different ways that silencers can be detected (or inferred), and it is not our goal to provide a comprehensive list here. However, we have added a specific example as suggested by the Reviewer at lines 172-177 to show how REDfly captures dual enhancer/silencer elements. (There are not very many of these in REDfly at the moment, but we expect the number will increase as we get caught up on curation.) Hopefully the addition of this example will clear up any confusion.
Reviewer 3 Report
In this manuscript, Keranen et al. describe the last updates of the database REDfly, a curated resource listing the regulatory elements of the Drosophila genome. The resource includes information on experimentally verified cis-regulatory modules and transcription factor binding sites. The authors first explain the utility of the database, giving some examples of usage by the scientific community, then they explain how the data are curated and annotated, specifying that the database also contains information on other insect species. At last, they give few examples explaining how to use the database to address specific questions. The database is useful and of interest for the community. The manuscript is well written and informative. I have few minor comments: I find the “use cases” maybe too detailed and if possible, I would add some information of how to add/submit new data to the database.
Author Response
We thank the reviewer for their comments. In response to the second comment, regarding adding/submitting new data, we have included a short paragraph at the end of the manuscript to address this point. In brief, all data submission is performed by our biocuration team; there is currently no mechanism for community data submission. However, we emphasize that we are happy to work with researchers to ensure that their data are rapidly and accurately entered into the database.
With respect to the first point, that the use cases are too detailed, we regret that the reviewer finds this so. However, in the absence of specific suggestions, we are reluctant to remove detail that we believe other, perhaps less savvy, users might find instructive. We note that the other referees commented that the use cases were a positive feature of the paper. Therefore, we have respectfully declined to alter that part of the manuscript.